# Quantitative Analysis of Seasonality and the Impact of COVID-19 on Tourists' Use of Urban Green Space in Okinawa: An ARIMA Modeling Approach Using Web Review Data

**Ruochen Yang** [1], **Kun Liu** [1], **Chang Su** [2], **Shiro Takeda** [1], **Junhua Zhang** [1] and **Shuhao Liu** [1,*]

1 Graduate School of Horticulture, Chiba University, Chiba 271-8510, Japan; 21hd0501@student.gs.chiba-u.jp (R.Y.); 22hd0501@student.gs.chiba-u.jp (K.L.); st@chiba-u.jp (S.T.); zhang@faculty.chiba-u.jp (J.Z.)

2 School of Architecture and Urban Planning, Huazhong University of Science and Technology, Wuhan 430074, China; suchang_la@hust.edu.cn

* Correspondence: 21hd0503@student.gs.chiba-u.jp; Tel.: +81-070-4331-3937

**Abstract:** We employed publicly available user-generated content (UGC) data from the website Tripadvisor and developed an autoregressive integrated moving average (ARIMA) model using the R language to analyze the seasonality of the use of urban green space (UGS) in Okinawa under normal conditions and during the COVID-19 pandemic. The seasonality of the use of ocean-area UGS is primarily influenced by climatic factors, with the peak season occurring from April to October and the off-peak season from November to March. Conversely, the seasonality of the use of non-ocean-area UGS remains fairly stable throughout the year, with a relatively high number of visitors in January and May. The outbreak of the COVID-19 pandemic greatly impacted visitor enthusiasm for travel, resulting in significantly fewer actual postings compared with predictions. During the outbreak, use of ocean-area UGS was severely restricted, resulting in even fewer postings and a negative correlation with the number of new cases. In contrast, for non-ocean-area UGS, a positive correlation was observed between the change in postings and the number of new cases. We offer several suggestions to develop UGS management in Okinawa, considering the opportunity for a period of recovery for the tourism industry.

**Keywords:** tourism; urban green space; COVID-19; Okinawa; ARIMA

## 1. Introduction

The Okinawa region in the southern part of Japan is a popular tourist destination known for its natural and cultural landscapes that showcase the maritime area. The region comprises the main island of Okinawa and numerous outlying islands; it draws many tourists, who visit for leisure and sightseeing and to experience Okinawa's unique culture [1]. As tourism has been the most important economic pillar of the region for many years, the development of tourism has significantly boosted the local economy, generating numerous jobs and substantial revenue [2].

The Okinawa region has 11 cities, including Naha, Ishigaki, and Ginowan, which are home to many local residents and are popular among tourists because of their easy accessibility to transportation and amenities (Figure 1) [3]. The most common areas of access for local residents and tourists fall under the category of urban green space (UGS), such as beaches, bays, parks, and traditional gardens in or around cities. These locales are characterized by their diversity, relative accessibility, and salient geographic features [4]. Scholarly studies and government reports indicate that Okinawa is highly regarded for its unique culture, cuisine, and historical sites. The beautiful maritime landscape and abundant marine activities are the primary reasons for which most tourists visit Okinawa and are the biggest selling points for tourism [5–7]. Owing to the special geographic environment,

Okinawa's UGS can be broadly classified into two sub-dimensions: ocean-area UGS and non-ocean-area UGS (Tables A1 and A2). Their common landscapes and support services play a vital role in shaping tourists' impressions and local quality of life [8].

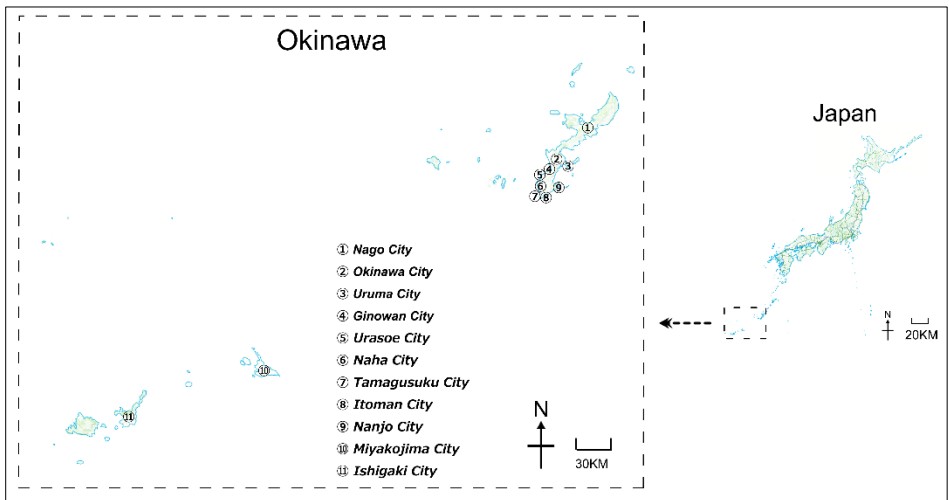

**Figure 1.** Location of the 11 cities in Okinawa.

Seasonal variability is a universal socio-economic phenomenon and one of the most critical aspects of the growth of the tourism sector [9]. Seasonality in tourism is influenced by both natural and man-made factors, the former being related to climatic variations and seasonal changes, and the latter being tied to a combination of social elements, such as customs and culture or unexpected events that directly or indirectly lead to systematic, cyclical, or irregular changes in various tourism indicators [10]. By understanding the seasonal patterns of tourists' use of Okinawa's UGS, we can better understand their behavioral patterns, and city managers can anticipate the flow of tourists in different types of areas at different times to deploy the necessary staff in advance, allocate materials and equipment appropriately, and devise more scientific and rational tourism development plans to better meet the needs of tourists and improve the efficiency of urban management. The results of this study could enable city managers to enhance Okinawa's image and reputation as a tourist destination, and thereby promote the tourism economy.

However, because of the outbreak of COVID-19, the Japanese government implemented stringent entry restrictions and quarantine measures to contain the spread of the virus. Combined with other components such as the temporary closure of public places, restrictions on business hours, and safety concerns for tourists visiting public sites, Okinawa's tourism industry suffered an unprecedented blow. In 2020 alone, the year the outbreak began, Okinawa's tourism revenue plummeted by 64% [11]. Consequently, most tourists cancelled their planned trips to Okinawa, leading to a significant decrease in the number of visitors. The use of UGS also declined during this period, and some activities and services were either postponed or cancelled, making it impossible for visitors to enjoy their trips as planned. This situation may have affected their use of UGS to a great extent [12].

With the increasing availability of vaccines and growing awareness of COVID-19, many countries are now gradually easing their travel restrictions. The Japanese government is expected to fully lift its strict policies by the summer of 2023, leading to a potential surge in the number of tourists [13]. However, to ensure a sustainable recovery of the tourism sector, a comprehensive approach, by reviewing past experiences and addressing new challenges, is required. This includes analyzing both actual and predictive value data to gain a better understanding of visitor behavior during the outbreak prevention and control period. This will provide a scientific reference for future decisions in the tourism industry [14].

Time-series modeling using past data and applying mathematical rules is currently the most popular method for quantitatively forecasting tourism demand [15]. Qin Hongyao [16] used tourism flight, traffic, and tourist data from a tourist city as research material to discuss the predictive effects of three methods—exponential smoothing, the seasonal autoregressive moving average (SARIMA), and the Elman artificial neural network technique—on the economic impacts of tourism in tourism demand forecasting. Using annual tourism industry statistics published in the China Statistical Yearbook of 1994–2015, Liu Ruyu [17] conducted a time-series analysis and constructed an autoregressive integrated moving average (ARIMA) model to forecast the number of domestic tourists in China for the next three years. Park [18] compared the correlation between news media topics in China and the US and the number of tourists traveling to Hong Kong from both countries and employed a SARIMA with exogenous factors (SARIMAX) model to forecast tourism demand from selected news topics and arrivals of tourists. In analyzing the impact of unexpected events, such as the COVID-19 outbreak on tourism, the analytical approach of annual data tends to ignore seasonal, cyclical, and stochastic factors in tourism, and, therefore, requires the use of monthly data with a high temporal resolution to provide a more comprehensive, in-depth understanding of the impact [19]. Laeeq Razzak [20] selected a dataset of international tourist arrivals in Thailand at the beginning of the COVID-19 outbreak and built an ARIMA model based on a four-step B-J method to predict the extent of the future recovery of Thailand's tourism economy. Prilistya [21] used web search trends as a dataset to predict the impact of the COVID-19 outbreak on the number of tourists traveling to Indonesia. However, the impact of outbreaks often radiates regionally, and the inherent patterns are difficult to capture simply by examining a region as a whole; hence, we need a comparative study with a control group. Furthermore, while the above studies are helpful in understanding the impact of emergencies on the overall tourism industry, businesses, and destinations, they neglect to focus on tourists, whose feelings during an emergency are more real, urgent, and accurate than after it has occurred. Thus, when studying the impact of an outbreak on tourism, observations and analysis should be centered on tourists [22]. It is worth exploring tourists' potential willingness to travel during an outbreak and whether the intensity of this willingness is different compared to normal times. By investigating tourists, we can better understand the impact of an outbreak on the tourism industry and provide a reference point for the future development of appropriate targeted measures.

User-generated content (UGC), which has increasingly been considered a trusted form of information dissemination in recent years, can be viewed as a data source. UGC has timeliness, diversity, and wide regional coverage compared with traditional data, and is more advantageous in capturing individual visitor behavior, predicting passenger flow trends, and conducting in-depth research on potential intentions [23]. Recent studies using UGC to explore the use of UGS during the COVID-19 pandemic include the study by Jato-Espino [24], who collated data about on-site use to investigate the beneficial social effects provided by green infrastructure during a strict blockade in Spain. Zhu [25] used correlation analysis based on Sina Weibo check-in data to assess the use of UGS in Beijing during COVID-19. Notwithstanding the above research, studies predicting changes in tourists' demand for UGS by collating UGC as a dataset are scarce. After careful consideration, we harnessed publicly available UGC from a travel review website (Tripadvisor) as a data source. We collected data on the number of visitor postings for ocean-area UGS and non-ocean-area UGS in the Okinawa region for a quantitative analysis using a time-series model.

## 2. Methods

### 2.1. Study Objects

We collected data on attraction sites across 11 cities in the Okinawa region, including Naha, Ginowan, Ishigaki, Urasoe, Nago, Itoman, Okinawa City, Tamagusuku, Uruma, Miyakojima, and Nanjo, from Tripadvisor, the world's largest travel review site. Tripadvisor has accumulated a vast amount of UGC from tourists worldwide and is widely used in Japan, where it is the most popular travel website in terms of user activity [26]. Tripadvisor

has also partnered with the Japan Tourism Association in recent years, making its data an industry standard and a reference for content, destination marketing, and trends in Japan's tourism industry [27].

Furthermore, official statistics on the number of visitors are only available for a few well-known attractions in the Okinawa region. For many free attractions that do not require tickets, official statistics may not provide comprehensive coverage. However, by utilizing the UGC data from Tripadvisor, we included 224 attractions in this study, most of which would be difficult to cover with official visitor statistics. This approach enabled us to indirectly assess visitor utilization trends for these attractions, which is a more advantageous method because it has a broader coverage of the study area compared to relying solely on official visitor count statistics. However, we used Tripadvisor as the sole data source for this study to prevent statistical bias arising from variations in the number of users across multiple travel websites and duplication of posts by the same user on different platforms [28,29]. This approach ensured that our data remained consistent and reliable.

### 2.2. Data Sources

The data sources included the following: (1) postings on Tripadvisor by tourists who matched the criteria for participants in the study; (2) statistics on the number of tourist arrivals in Okinawa published on the official website of the Okinawan government [30]; and (3) data on new cases of COVID-19 in Okinawa published by the Ministry of Health, Labour, and Welfare [31].

We obtained data from tourists' postings on Tripadvisor using a web crawling method. First, based on the Python 3.7 language, we employed the Scrapy crawler framework to view the source code of Tripadvisor to find the target data location that matched the research object, and we analyzed the webpage's source code structure. Subsequently, we used the Python library requests and the Beautiful Soup library to document the webpage for data collection, and we used the PyMongo library to read the data cyclically. We divided and stored the page number intervals in the database with the help of the range function and set a cyclic threshold for crawling. Finally, we de-weighted and filtered the acquired data; we generated an Excel table to obtain the raw data and carried out reasonable data cleaning and processing for future research [32]. All the above data are public information that do not involve personal privacy and will not affect normal use of the website.

### 2.3. Modeling the ARIMA

In this study, we took into consideration Okinawa's unique maritime nature and official tourism reports. We distinguished the collected data on the basis of two categories—ocean-area UGS and non-ocean-area UGS—based on the attraction information provided in the original data and with reference to previous studies on tourism demand in Okinawa, as well as the composition of Okinawa's landscape. To predict trends in visitor use, we employed the R language (hereafter called R) to develop separate seasonal ARIMA models for different types of UGS, assuming they were not affected by the pandemic.

The ARIMA model is a very accurate method among time-series forecasting models with high forecasting accuracy; it is suitable for solving linear model forecasting problems. The trend of visitor usage of UGS (which is highly influenced by irregular shocks) is long-term, seasonal, and non-stationary. Forecasting using the ARIMA model can reduce forecasting errors. The basic principle of the ARIMA model is to convert a time series into a stationary time series and fit a forecasting model via autoregression and the moving average of this series. The formula of the ARIMA model comprises three parts: (1) autoregression (AR); (2) the moving average (MA); and (3) the difference (differential treatment of the data). The general form of the ARIMA model can be expressed as $p$ (the number of AR terms), $d$ (the number of differences), and $q$ (the number of MA terms) [33].

The process of building the ARIMA model using R included: (1) loading the time-series data of the number of visitor postings and using the read.csv function to read the CSV file (which stored the number of visitor postings) in R; (2) plotting the time series of the data,

observing the characteristics of changes in the data, and determining the smooth white noise series that the data are in; (3) plotting the autocorrelation function (ACF) and partial ACF (PACF) graphs and observing them to determine the order of the ARIMA model; (4) fitting the ARIMA model using the ARIMA function in R and using the forecast function to make a 24-period forecast; (5) using the ts.diag function in R to test the significance of the model and the box.test function to test the independence of the model residuals to verify that the ARIMA model fits the data series well; and (6) using the fitted ARIMA model to predict visitor usage assuming no impact from the pandemic [34].

Subsequently, we compared proportional changes between the actual values during the pandemic prevention and control (PPC) period and the predictive values obtained through the ARIMA model. We plotted a line graph of the proportional changes to evaluate the magnitude of changes in the pandemic's impact on past visitor use based on seasonality. Finally, we used the trend of new COVID-19 cases in Okinawa during the same period to measure the change in the trend of tourists' use of Okinawa's UGS before and after the pandemic. The goal of these analyses was to elucidate the intrinsic patterns and shifting trends in Okinawa's tourism industry. This information will be valuable for future crisis response and resource management regulations in the tourism recovery process.

## 3. Results

### 3.1. Results of Crawling the Research Data

The collected visitor data included information on travel time (year and month), attraction nomenclature, and attraction type. After removing invalid data on advertising content and missing information, we obtained 20,573 pieces of valid raw data, of which the earliest piece dated back to June 2008; further, to ensure uniformity in the time-seasonal variables of the data, we took the complete annual data after January 2009. As shown in Table 1, the posting numbers from 2009 to 2013 among the raw data show a small base and a multiplicative increase in the overall number of visitors, with little change in official statistics; thus, we can deduce that this time period is in the accumulation phase of user enthusiasm on the website, and the posting numbers exceeded 2000 for the first time in 2014. Subsequent data did not exhibit significant changes before the outbreak. In addition, the Japanese government announced the first case of COVID-19 in Okinawa on January 15 2020, and issued several states of emergency, restricted commercial activities, and imposed a strict quarantine entry policy. After 2022, the Japanese government gradually stopped issuing states of emergency and lifted the quarantine entry policy for some countries and regions [35]. Thus, the period from early 2020 to early 2022 can be considered the time when the Japanese government implemented a strict PPC policy.

**Table 1.** Volume of raw data and visitor statistics between 2009 and 2019.

| Years | Volume of Data | Number of Tourists |
|-------|----------------|--------------------|
| 2009 | 87 | 5,690,000 |
| 2010 | 221 | 5,717,900 |
| 2011 | 284 | 5,528,000 |
| 2012 | 575 | 5,924,700 |
| 2013 | 995 | 6,583,000 |
| 2014 | 2034 | 7,169,900 |
| 2015 | 3187 | 7,936,300 |
| 2016 | 3442 | 8,769,200 |
| 2017 | 3398 | 9,579,900 |
| 2018 | 2942 | 10,004,300 |
| 2019 | 2310 | 10,163,900 |

In summary, we selected the data for the study from the pre-outbreak period (January 2014–December 2019) and the strict PPC period (January 2020–December 2021); we obtained 17,313 entries for the pre-outbreak period, and 1098 for the strict PPC period.

### 3.2. Results of Collating the Time-Series Dataset

We collated the time-series dataset of the number of visitor posts (Table 2). In the pre-pandemic period, 11,070 posts were about ocean-area UGS and 6243 posts were about non-ocean-area UGS. In the strict PPC period, 565 posts were on ocean-area UGS and 533 posts were on non-ocean-area UGS.

**Table 2.** Time series of the posting numbers about ocean-area UGS and non-ocean-area UGS.

| Time (Month and Year) | Ocean-Area UGS | Non-Ocean-Area UGS |
|---|---|---|
| January 2014 | 68 | 57 |
| February 2014 | 56 | 58 |
| March 2014 | 111 | 57 |
| April 2014 | 108 | 69 |
| May 2014 | 104 | 54 |
| June 2014 | 105 | 44 |
| July 2014 | 170 | 56 |
| August 2014 | 175 | 50 |
| September 2014 | 170 | 58 |
| October 2014 | 100 | 52 |
| November 2014 | 88 | 68 |
| December 2014 | 84 | 72 |
| January 2015 | 104 | 95 |
| February 2015 | 96 | 93 |
| March 2015 | 105 | 83 |
| April 2015 | 161 | 98 |
| May 2015 | 181 | 103 |
| June 2015 | 205 | 91 |
| July 2015 | 265 | 97 |
| August 2015 | 236 | 86 |
| September 2015 | 214 | 88 |
| October 2015 | 213 | 110 |
| November 2015 | 122 | 78 |
| December 2015 | 135 | 128 |
| January 2016 | 130 | 133 |
| February 2016 | 129 | 107 |
| March 2016 | 156 | 106 |
| April 2016 | 253 | 116 |
| May 2016 | 225 | 110 |
| June 2016 | 195 | 84 |
| July 2016 | 286 | 112 |
| August 2016 | 236 | 94 |

**Table 2.** *Cont.*

| Time (Month and Year) | Ocean-Area UGS | Non-Ocean-Area UGS |
|---|---|---|
| September 2016 | 201 | 74 |
| October 2016 | 192 | 85 |
| November 2016 | 115 | 73 |
| December 2016 | 124 | 106 |
| January 2017 | 160 | 126 |
| February 2017 | 104 | 86 |
| March 2017 | 136 | 124 |
| April 2017 | 261 | 147 |
| May 2017 | 214 | 92 |
| June 2017 | 205 | 84 |
| July 2017 | 284 | 89 |
| August 2017 | 237 | 81 |
| September 2017 | 166 | 78 |
| October 2017 | 267 | 99 |
| November 2017 | 97 | 71 |
| December 2017 | 104 | 86 |
| January 2018 | 112 | 119 |
| February 2018 | 104 | 108 |
| March 2018 | 123 | 110 |
| April 2018 | 160 | 120 |
| May 2018 | 173 | 94 |
| June 2018 | 156 | 67 |
| July 2018 | 182 | 77 |
| August 2018 | 234 | 109 |
| September 2018 | 160 | 92 |
| October 2018 | 148 | 79 |
| November 2018 | 132 | 86 |
| December 2018 | 103 | 94 |
| January 2019 | 123 | 106 |
| February 2019 | 97 | 80 |
| March 2019 | 121 | 86 |
| April 2019 | 123 | 83 |
| May 2019 | 141 | 95 |
| June 2019 | 150 | 61 |
| July 2019 | 164 | 62 |
| August 2019 | 125 | 57 |
| September 2019 | 97 | 62 |
| October 2019 | 134 | 68 |
| November 2019 | 88 | 52 |
| December 2019 | 67 | 68 |

### 3.3. Results of the Time-Series Modeling

We wished to build on the historical time-series data from before the outbreak to predict seasonality of use assuming no impact of the pandemic. We took the following steps based on Box–Jenkins' approach [36]. First, we analyzed the time-series data, including the pre-processing of the data, making graphs, and determining the trend, periodicity, and seasonality of the data. Subsequently, depending on the characteristics of the historical data, we selected two to three potential models for model identification, as well as for the strength of correlation between the predictor variables and the explanatory variables [37]. Following that, we calculated the model's parameters and assessed the model's fit. Finally, in the process of estimating and testing the models, if there were deficiencies, we carried out model optimization and improvement and applied the trained model data for comparative analysis.

#### 3.3.1. An ARIMA Model for Ocean-Area UGS

(a)  Modeling

First, we plotted the time-series data of ocean-area UGS before the outbreak, as seen in Figure 2. From the time-series plot, we can deduce that the data series has a certain increasing trend and a large cyclical feature; therefore, we deemed the series to be non-stationary. Since the model is a non-stationary series with seasonality, we calculated the differences in the data series according to the first-order 12 steps; the differenced time series is plotted in Figure 3. The data series of ocean-area UGS before the outbreak fluctuates around the value of 0 after the first order of 12-step differencing, and there is no obvious trend.

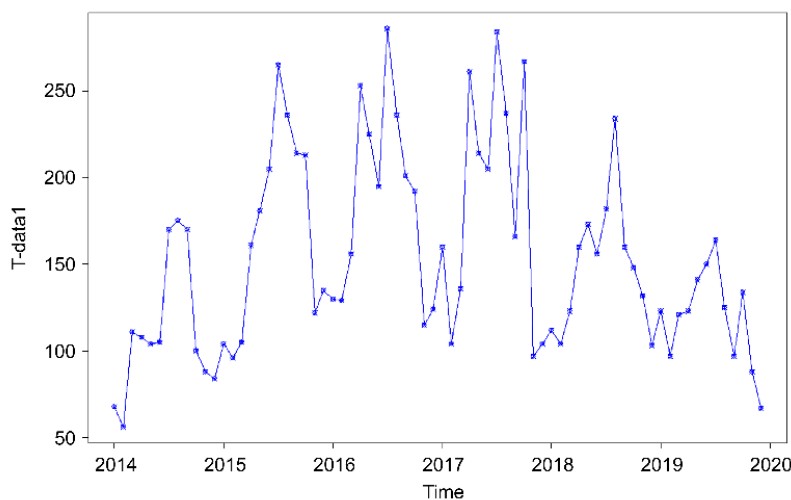

**Figure 2.** Time-series graph of data on the posting numbers in ocean-area UGS before the outbreak.

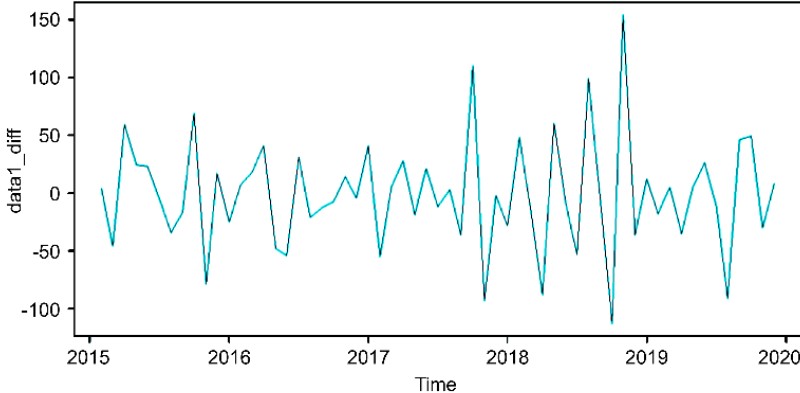

**Figure 3.** Time-series plot of the first-order difference regarding the posting numbers for ocean-area UGS before the outbreak.

To further determine the smoothness of the differenced series, we performed a test for smoothness and a test for the pure randomness of the series on the differenced series. We carried out a unit root test on the dataset using the adf.test function in R to determine whether the dataset was smooth. We employed the augmented Dickey–Fuller (ADF) test to set up the test results for three hypothetical cases: Type 1: no drift, no trend, where the dataset is tested for smoothness without considering drift or trend; Type 2: with drift but no trend, where the dataset is tested for smoothness with drift, but without considering trend; and Type 3: with drift and trend, where the dataset is tested for smoothness with both drift and drift and trend, which determines whether the dataset is smooth. For each hypothesis, we found that the *p*-value of the ADF test statistic for the ocean-area UGS data series after first-order differencing was less than the 0.05 level of significance; thus, we were able to determine that the series, after first-order 12-step differencing of the data, achieved smoothness.

Subsequently, we used the Ljung–Box (LB) test to determine the pure randomness of the differenced series. This series at order 6 had a statistic of $X^2 = 19.981$, df = 6, $p = 0.002791$; at order 12, the series had a statistic of $X^2 = 35.709$, df = 12, $p = 0.0003608$. Thus, this series at orders 6 and 12 delayed the *p*-values of the LB statistics of this series, which are less than the significance level of 0.05. We were able to establish that the series of ocean-area UGS data before the outbreak after first-order 12-step differencing were all smooth white noise series; thus, we rejected the original hypothesis of pure randomness. Therefore, we can consider this differenced series to be non-random.

We then plotted the autocorrelation and partial autocorrelation plots of the first-order differenced series (Figure 4) to examine the characteristics of the autocorrelation and partial autocorrelation coefficients up to order 12 for this data series. This allowed us to determine the model's fit. We can see that the autocorrelation coefficients up to order 12 are truncated and the partial autocorrelation coefficients are not truncated, so we attempted to extract the short-term autocorrelation information of the differenced series using the ARIMA (0,1) model. Given its possible seasonal autocorrelation characteristics, this time series investigates the characteristics of the delayed 12th- and 24th-order autocorrelation coefficients in terms of period length and the partial autocorrelation coefficients. The autocorrelation plot shows that the delayed 12th-order autocorrelation coefficient is significantly non-zero, but the delayed 24th-order autocorrelation coefficient falls into the two-times standard deviation range. The partial autocorrelation plot indicates that the partial autocorrelation coefficients for both the delayed 12th and 24th orders are significantly non-zero. Thus, we can assume that the seasonal autocorrelation is characterized by a truncated tail of autocorrelation coefficients and a trailing tail of partial autocorrelation coefficients when the ARIMA (0,1) model, with a 12-step period, extracts the seasonal autocorrelation information of the differenced series. Combining the previous differencing information, we chose the fitted product model as ARIMA (0,1,1) × (0,1,1) [12].

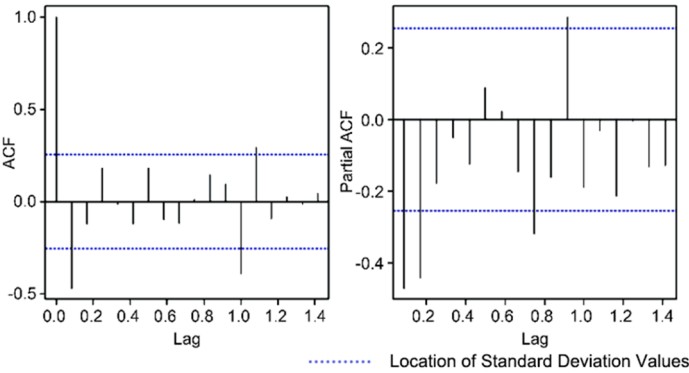

**Figure 4.** Autocorrelograms and partial autocorrelograms of the first-order difference posterior series of posting numbers for ocean-area UGS before the outbreak.

In sum, we used the ARIMA (0,1,1) × (0,1,1) [12] model to fit the ocean-area UGS data series from January 2014 to December 2019. We harnessed the forecast function to make a 24-period forecast, and the predictive values of ocean-area UGS data from January 2020 to December 2021 under the assumption of no impact of the pandemic (see Table 3; the forecast effect is presented in Figure 5).

**Table 3.** Predictive value for ocean-area UGS from January 2020 to December 2021, assuming no impact of the pandemic (data from the ARIMA model using R).

| Time (Month and Year) | Predictive Value (Ocean-Area UGS) |
| --- | --- |
| January 2020 | 78.08404 |
| February 2020 | 57.38566 |
| March 2020 | 83.26087 |
| April 2020 | 129.76301 |
| May 2020 | 128.58841 |
| June 2020 | 124.91645 |
| July 2020 | 171.0254 |
| August 2020 | 154.29757 |
| September 2020 | 111.09483 |
| October 2020 | 128.38786 |
| November 2020 | 63.48787 |
| December 2020 | 54.03563 |
| January 2021 | 68.76185 |
| February 2021 | 48.06347 |
| March 2021 | 73.93868 |
| April 2021 | 120.44082 |
| May 2021 | 119.26622 |
| June 2021 | 115.59426 |
| July 2021 | 161.70321 |
| August 2021 | 144.97538 |
| September 2021 | 101.77264 |
| October 2021 | 119.06566 |
| November 2021 | 54.16567 |
| December 2021 | 44.71344 |

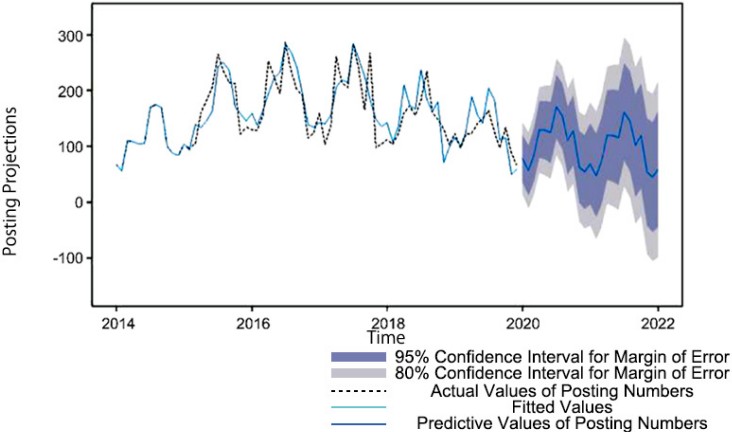

**Figure 5.** Predictive values for ocean-area UGS from January 2020 to December 2021, assuming no impact of the pandemic (data from the ARIMA model using R).

(b) Testing the Model

First, we performed the significance test using the ts.diag function in R. The residual series are white noise series, indicating that the model fits well and extracts sufficient information about the series. In addition, a joint plot of the fitted and observed values of the series indicates that the ARIMA model fits the series well (Figure 6). Subsequently, to determine whether the prediction error was normally distributed with a 0 mean and constant variance, we performed the independence test of the model's residual using the box.test function. The residual plot is depicted in Figure 7. Upon constructing the plot ForecastErrors function to convert the residual into a normal distribution plot (Figure 8), we found that it passed the normality test. Subsequently, we built the autocorrelation plot of residuals (Figure 9) and performed the LB test to determine whether the residual series was autocorrelated. The outcome of $p = 0.4714$ was greater than the 0.05 level of significance, and the original hypothesis could not be rejected; thus, the residual series cannot be considered autocorrelated. All tests were passed, so we can assume that the ARIMA model is a good fit.

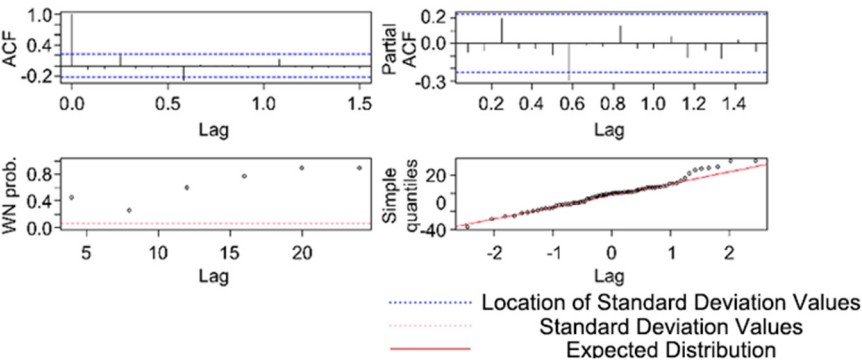

**Figure 6.** Joint plot of series fits and series observations of predictive values for ocean-area UGS.

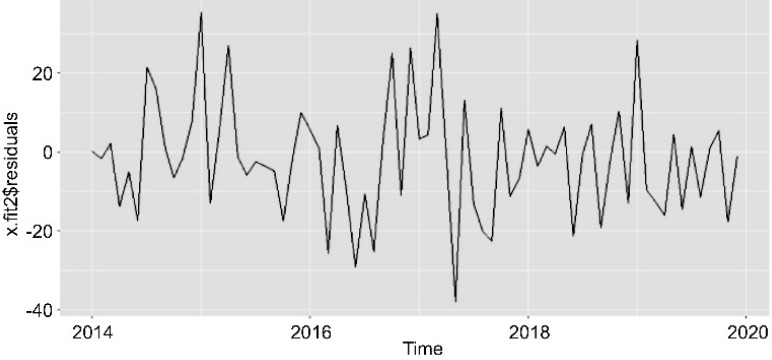

**Figure 7.** Residual plots from the ARIMA model of ocean-area UGS.

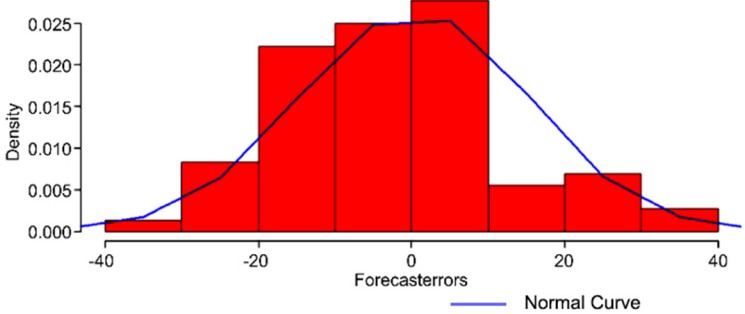

**Figure 8.** Normal distribution transformed from the residuals of the ARIMA model of ocean-area UGS.

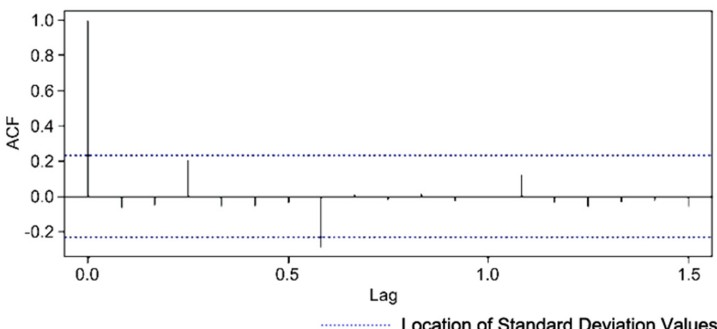

**Figure 9.** Autocorrelogram of the residuals from the ARIMA model of ocean-area UGS.

### 3.3.2. ARIMA Model for Non-Ocean-Area UGS

We carried out the same process with the ARIMA model for the pre-outbreak ocean-area UGS; we were able to test the results of the model that we built, which proved to be a good fit. We plotted the time-series data of ocean-area UGS before the outbreak as shown in Figure 10. By examining the autocorrelation and partial autocorrelation plot features (Figure 11), we used the ARIMA (0,1,1) × (0,1,1) [12] model to fit the non-ocean-area UGS data series from January 2014 to December 2019. We fitted the ocean-area UGS data series and the predictive values of non-ocean-area UGS data from January 2020 to December 2021 under the assumption of no impact from the pandemic (see Table 4). The prediction effect is shown in Figure 12.

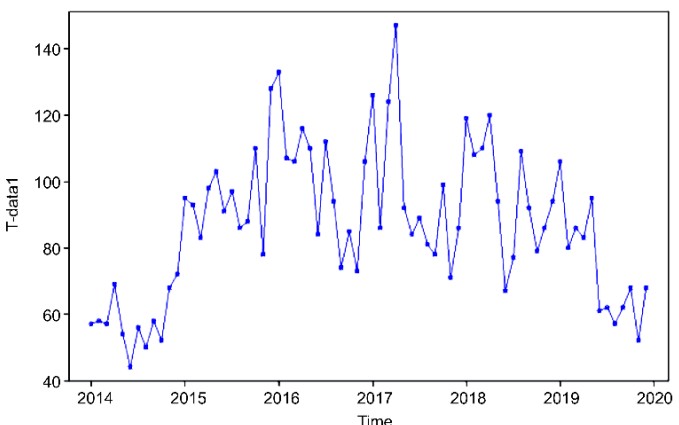

**Figure 10.** Time-series graph of data on the posting numbers in non-ocean-area UGS before the outbreak.

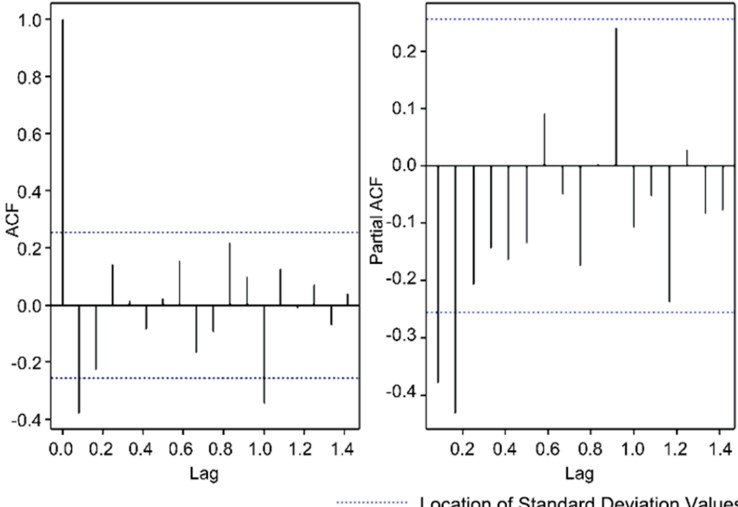

**Figure 11.** Autocorrelograms and partial autocorrelograms of the first-order difference posterior series of posting counts for non-ocean-area UGS before the outbreak.

**Table 4.** Predictive value for non-ocean-area UGS from January 2020 to December 2021, assuming no impact of the pandemic (data from the ARIMA model using R).

| Time (Month and Year) | Predictive Value (Non-Ocean-Area UGS) |
| --- | --- |
| January 2020 | 69.36227 |
| February 2020 | 62.42041 |
| March 2020 | 60.41193 |
| April 2020 | 57.79053 |
| May 2020 | 66.66478 |
| June 2020 | 51.83195 |
| July 2020 | 53.09759 |
| August 2020 | 53.94552 |
| September 2020 | 54.32503 |
| October 2020 | 53.93627 |
| November 2020 | 50.57369 |
| December 2020 | 57.13684 |
| January 2021 | 55.19529 |
| February 2021 | 50.41964 |
| March 2021 | 50.97723 |
| April 2021 | 49.22835 |
| May 2021 | 54.82956 |
| June 2021 | 49.18297 |
| July 2021 | 49.10758 |
| August 2021 | 45.08598 |
| September 2021 | 47.14309 |
| October 2021 | 49.37000 |
| November 2021 | 45.56541 |
| December 2021 | 48.69557 |

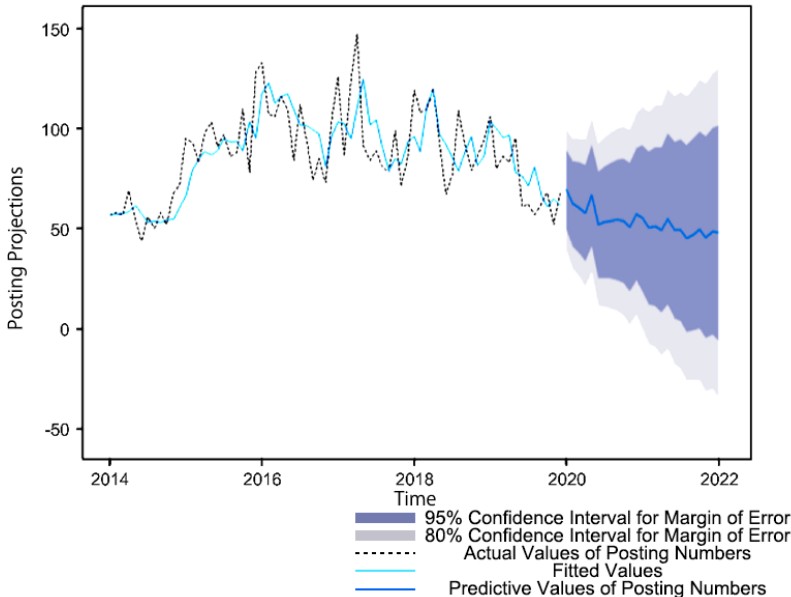

**Figure 12.** Predictive values for non-ocean-area UGS from January 2020 to December 2021, assuming no impact of the pandemic (data from the ARIMA model using R).

### 3.3.3. Magnitude of Change in Proportional Relationships

We separately analyzed the proportional relationship between the actual-values date series after the COVID-19 outbreak (see Table 5) and the predictive-values data series. This helped us avoid errors caused by differences in the databases used, and enabled us to determine the magnitude of change that occurred after the outbreak, grounded in the seasonal conditions used in the past. Table 6 presents the proportional relationship between the actual and predictive values of posting numbers post-outbreak for UGS in ocean and non-ocean areas.

**Table 5.** Actual-values data series of ocean and non-ocean-area UGS after the outbreak.

| Time (Month and Year) | Actual Value (Ocean-Area UGS) | Actual Value (Non-Ocean-Area UGS) |
|---|---|---|
| January 2020 | 62 | 65 |
| February 2020 | 66 | 48 |
| March 2020 | 72 | 46 |
| April 2020 | 12 | 13 |
| May 2020 | 1 | 4 |
| June 2020 | 9 | 9 |
| July 2020 | 40 | 27 |
| August 2020 | 30 | 25 |
| September 2020 | 12 | 8 |
| October 2020 | 31 | 9 |
| November 2020 | 44 | 28 |
| December 2020 | 32 | 24 |
| January 2021 | 7 | 25 |
| February 2021 | 4 | 6 |
| March 2021 | 8 | 11 |
| April 2021 | 12 | 24 |
| May 2021 | 24 | 15 |
| June 2021 | 8 | 14 |
| July 2021 | 14 | 19 |
| August 2021 | 19 | 32 |
| September 2021 | 13 | 20 |
| October 2021 | 8 | 11 |
| November 2021 | 11 | 22 |
| December 2021 | 26 | 28 |

**Table 6.** Proportional relationship between the actual and predictive values.

| Time (Month and Year) | Proportion (Ocean-Area UGS) | Proportion (Non-Ocean-Area UGS) |
|---|---|---|
| January 2020 | 79.40% | 93.71% |
| February 2020 | 115.01% | 76.90% |
| March 2020 | 86.48% | 76.14% |
| April 2020 | 9.25% | 22.50% |

**Table 6.** *Cont.*

| Time (Month and Year) | Proportion (Ocean-Area UGS) | Proportion (Non-Ocean-Area UGS) |
|---|---|---|
| May 2020 | 0.78% | 6.00% |
| June 2020 | 7.20% | 17.36% |
| July 2020 | 23.39% | 50.85% |
| August 2020 | 19.44% | 46.34% |
| September 2020 | 10.80% | 14.73% |
| October 2020 | 24.15% | 16.69% |
| November 2020 | 69.30% | 55.36% |
| December 2020 | 59.22% | 42.00% |
| January 2021 | 10.18% | 45.29% |
| February 2021 | 8.32% | 11.90% |
| March 2021 | 10.82% | 21.58% |
| April 2021 | 9.96% | 48.75% |
| May 2021 | 20.12% | 27.36% |
| June 2021 | 6.92% | 28.47% |
| July 2021 | 8.66% | 38.69% |
| August 2021 | 13.11% | 70.98% |
| September 2021 | 12.77% | 42.42% |
| October 2021 | 6.72% | 22.28% |
| November 2021 | 20.31% | 48.28% |
| December 2021 | 58.15% | 57.50% |

## 4. Discussion

### 4.1. Ocean-Area UGS

#### 4.1.1. Exploiting Seasonality

We created a line graph to predict the posting values for ocean-area UGS, assuming no impact from the outbreak. The data series used in the graph was rooted in pre-outbreak data, as shown in Figure 13. The graph clearly illustrates the seasonal nature of visitor use of the area, with a noticeable cyclical pattern throughout the year. The peak season for visitors to ocean-area UGS occurs from April to October, while the off-peak season is from November to March. During the peak season, posting forecasts for April to June are almost identical, with a sharp increase in July, reaching the highest point. Posting forecasts are somewhat high in August and begin to decline significantly in September, with a slight increase in October. During the off-peak season, posting forecasts drop significantly in November compared to October, reaching the lowest point of the year in December. Posting forecasts gradually increase in January, reach a similar trough in February and December, and begin a sustained upward trend in March.

First, we must consider the seasonal impact of climate factors on the use of ocean areas. Let us take Naha, the largest city in Okinawa, as an example. In Naha, the temperature gradually rises above a comfortable 20 °C starting in April, allowing tourists to easily engage in various water activities. The rainy season begins in early May in Okinawa and ends in late June. Afterward, the Pacific high-pressure system covers the region, and there are often consecutive sunny days in July with less precipitation. This is also the period of the strongest ultraviolet radiation, which we claim is the main reason for the highest number of posting forecasts in that month. From August to early October, the area is susceptible to typhoons, and marine-related activities are restricted, gradually entering

the off-peak season for tourism. Starting in November, the average temperature begins to drop below 20 °C, and the duration and intensity of sunshine also decrease significantly. The average duration and intensity of sunshine from December to February are less than half of those in July. We assert that this is also the chief reason for the low number of posting forecasts in December and February. In March, the temperature gradually warms up. During this time, cold air from the Asian continent tends to form a cloud cover when passing over the warm East Sea, and sunlight is not abundant [38].

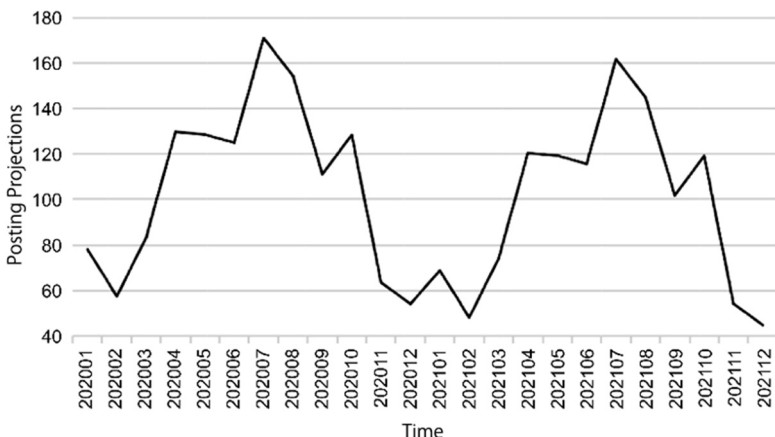

**Figure 13.** Line graph of predictive values for ocean-area UGS, assuming no impact of the pandemic.

Second, holiday factors have a certain influence on seasonal use. Tourists traveling with their families account for the majority of visitors to Okinawa [39]. Although early May is national Golden Week in Japan, when many people go on vacation, no significant increase was observed in predictive values in that month. By contrast, July is the start of summer vacation for Japanese students, and parents take their children outside to play, leading to a peak period for tourism in Okinawa, resulting in higher posting forecasts in July. When planning a travel itinerary, tourists usually consider the climate characteristics of the destination, making UGS near the ocean more popular during summer vacation. In contrast, such areas are not in the travel plans of most tourists during students' winter vacation from February to March. Additionally, most overseas visitors to Okinawa come from Greater China [28], where, in many places, people celebrate National Day in October. Meanwhile, the occurrence of typhoons ceases in October, and the average temperature can remain around a comfortable 25 °C for water activities. Therefore, climate and holiday factors may be common reasons for the small increase in predictive values in October. Although the climate in January is not suitable for water activities, it is the most important New Year's holiday in Japan, and the number of tourists rises significantly. Some tourists visit the area to admire the ocean views, which may be why the predictive values in January are relatively high during the off-peak season.

4.1.2. Impact of the Outbreak

We jointly mapped the status of the number of new cases in Okinawa and the proportional relationship between the actual and predictive values of posting numbers after the outbreak in ocean-area UGS, as shown in Figure 14. The analysis indicates that, at the start of the outbreak, the impact on the relative posting numbers was small and even exceeded the predictive value in February 2020. However, after April 2020, a significantly negative correlation was observed between the situation of new cases and the relative number of postings, resulting in a significant decline. We noted the most pronounced drop in the relative number of postings during the first small peak of the pandemic in April 2020. Although the relative number of postings fluctuated along with the second mini-peak of the outbreak in August 2020, it remained low and did not exceed a quarter of the predictive value until October 2020. As the number of new cases plateaued between September and

December 2020, the decline in the relative number of postings tapered off and reached 69.30% of the predicted value, indicating a short-lived trend of utilizing the "new normal" during this period. However, from January 2021 onward, four consecutive peaks in the number of new cases resulted in a continued downturn in the relative number of postings, with only a small recovery between the fourth and fifth peaks. As the number of new cases decreased after October 2021, the relative posting numbers began to rebound rapidly. Overall, the relative number of postings in ocean-area UGS shows a negative correlation with the number of new cases, and there is a small lag effect between changes. After a spike in new cases, the magnitude of the relative postings began to drop significantly.

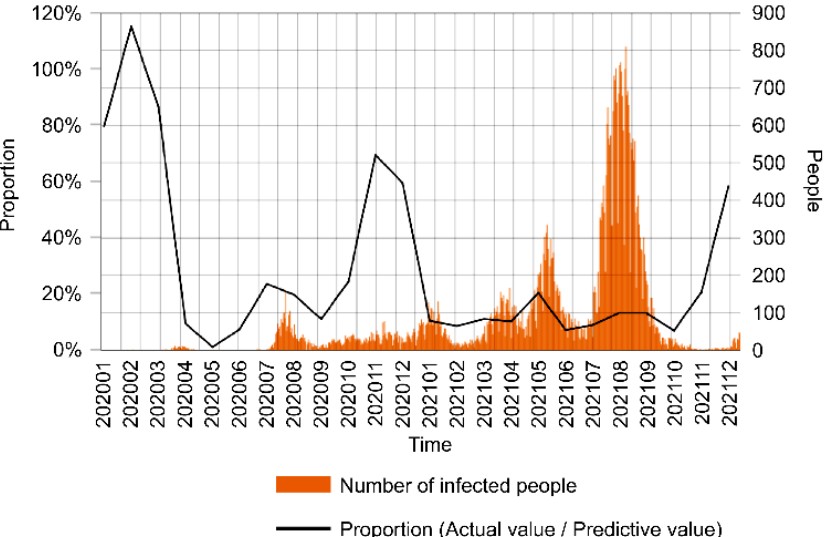

**Figure 14.** Joint mapping of the status of the number of new cases in Okinawa and the proportional relationship between the actual and predictive values of posting numbers after the outbreak in ocean-area UGS.

*4.2. Non-Ocean-Area UGS*

4.2.1. Exploiting Seasonality

According to the pre-outbreak data series, we plotted a line graph of predictive values for non-ocean-area UGS, which we assumed to be unaffected by the outbreak, as shown in Figure 15. The graph showed no clear trend of the seasonal use of these green spaces throughout the year, indicating that climate might not be a significant factor in determining visitors' tendencies to use such spaces. Furthermore, even before the outbreak, there was already a noticeable downward trend in online postings for this type of green space. Despite this, posting predictions for non-ocean-area UGS show relatively high numbers during the New Year's holidays in January and Golden Week in May, which are periods in the year when Japanese people go on vacation the most. Between November and March, when ocean-area UGS is in its off-peak season, posting forecasts remain low in November and start to increase significantly in December. This may be related to discounted accommodations and transport prices during this period [6], as well as the intensification of traditional cultural events in the Okinawa region close to the New Year and an increase in visitors seeking refuge from the winter weather [40], among other factors.

4.2.2. Impact of the Outbreak

A joint graph of the status of the number of new cases in Okinawa and the proportional relationship between the actual and predictive values following the outbreak in non-ocean-area UGS is shown in Figure 16. The overall decrease in the region is slightly less than that of ocean-area UGS, and the impact of the outbreak is somewhat small. The reduction in the relative number of postings was most pronounced when the first small peak of the pandemic occurred in April 2020. After the first outbreak, the relative number of postings

began to rise gradually, reaching 50.58% of the predictive value at one point during the second small peak of the outbreak in August 2020. However, after the number of new cases declined from September to October 2020, the relative number of postings in this type of green space dropped significantly to 16.69%. During the period of continued growth in the number of new cases from November 2020 to January 2021, the number of postings and the percentage of forecasts began to rise, reaching a maximum of 55.36% of all forecasts. The relative posting numbers then fluctuated positively according to the trends of increases and decreases in the number of new cases. During the highest number of new cases in August 2021, the relative number of postings reached the highest point after the first small peak of the pandemic, reaching 70.98% of the predictive value. The number of people rose and the relative number of postings increased again to exceed half in December 2021, reaching 57.50%. Overall, the relative number of postings in non-ocean-area UGS indicates a positive correlation with the number of new cases. The lag effect of the subsequent change was small; thus, once there was a peak in new cases, the magnitude of the proportion of postings began to expand compared to the predictive value.

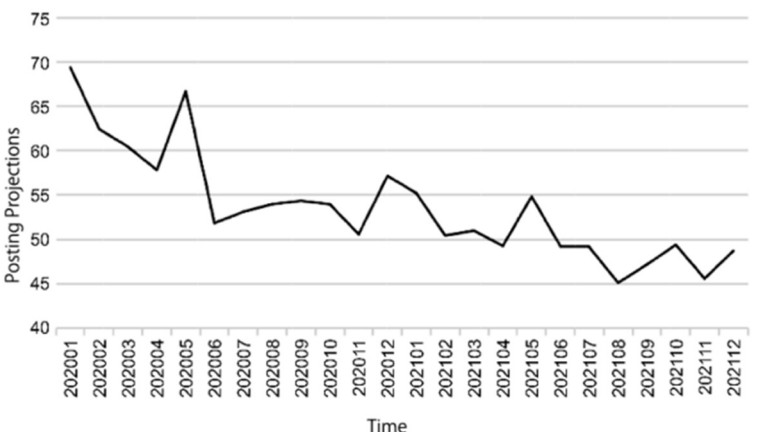

**Figure 15.** Line graph of predictive values for non-ocean-area UGS, assuming no impact of the pandemic.

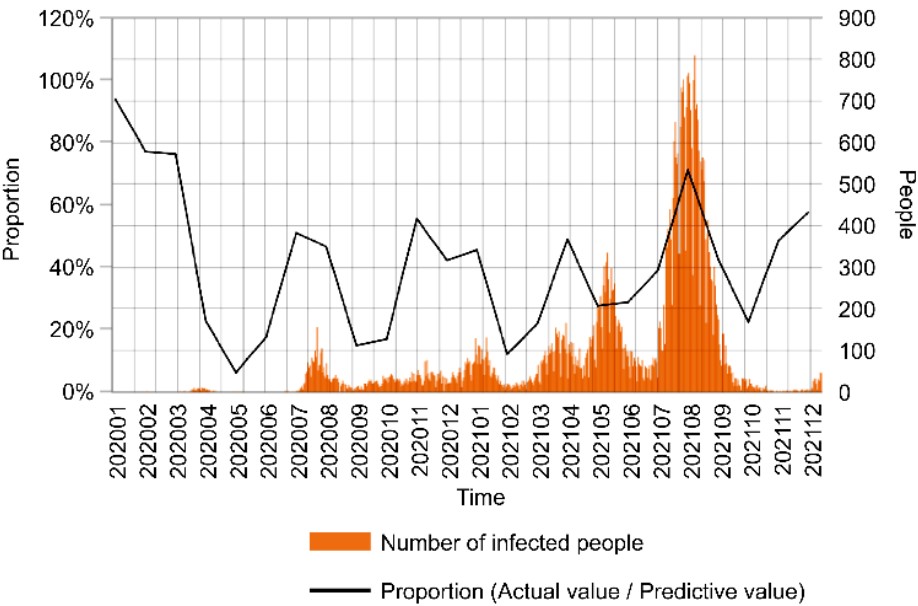

**Figure 16.** Joint mapping of the status of the number of new cases in Okinawa and the proportional relationship between the actual and predictive values of posting numbers after the outbreak in non-ocean-area UGS.

*4.3. Comparative Analysis*

The graph in Figure 17 shows a crossed line plot, illustrating the proportional relationship between the actual and predictive values of posting numbers for the two types of UGS (ocean-area UGS and non-ocean-area UGS). Before the outbreak, ocean-area UGS had been experiencing an upward trend in online posting popularity, while non-ocean-area UGS had been experiencing a downward trend. However, after the first small spike in the number of new cases, the relative posting numbers for both types of UGS declined significantly. This was due to many tourists having to cancel their planned trips because of various restrictions implemented to control the spread of the virus. Despite the overall decrease in the number of postings for both types of UGS, the magnitude of the decrease and the trend of the increase and decrease in new cases were quite different. This suggests that other factors may have also influenced the popularity of UGS during the pandemic.

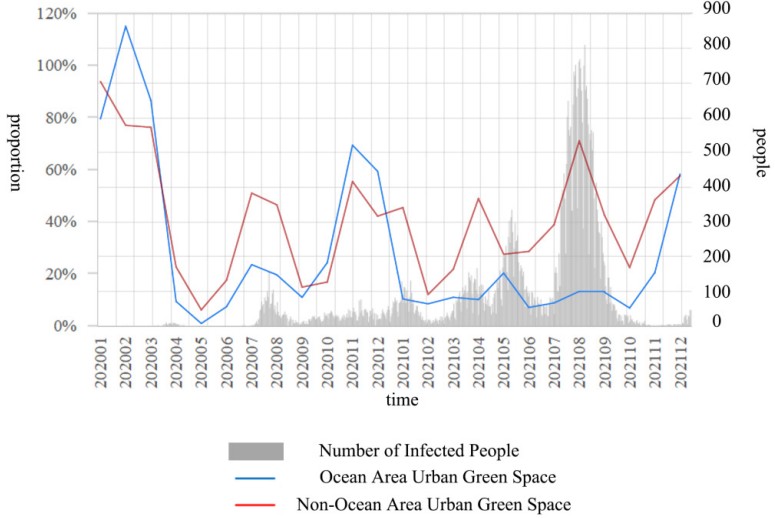

**Figure 17.** Crossed line graph of the proportional relationship between the actual and predictive values of posting numbers about ocean-area UGS and non-ocean-area UGS.

First, ocean-area UGS is popular among tourists, drawing many visitors. Notwithstanding, research has shown that, after an outbreak [41], there is a general perception of a greater risk of infection in popular, crowded, high-density areas, which visitors tend to avoid. At the same time, visitors have become more inclined to opt for spectator activities at a social distance, influenced by the publicity of the PPC policy. By contrast, the more interactive nature of marine recreational activities, as well as the government's tendency to restrict water activities by closing or suspending access during peak numbers of new cases, further reduced the number of tourist arrivals by making certain areas unavailable to visitors. This may have contributed to a greater reduction in the relative number of postings about ocean-area UGS.

Second, upon analyzing the trend in the number of new cases, we observed a negative correlation with the relative number of postings about ocean-area UGS. This means that, when the number of new cases began to show an increasing trend, the number of postings about ocean-area UGS fell significantly. By contrast, non-ocean-area UGS showed a positive correlation with the number of postings, with the relative posting numbers rising as new cases began to emerge. One possible explanation for this phenomenon is that, in the event of an increase in the number of new cases, tourists still wish to visit green spaces (such as beaches), even after they are closed. People may need to find alternative areas such as urban parks or traditional gardens. As most ocean-area UGS are located far from accommodations in the city center, people might prefer less crowded areas that are easy to reach. However, while non-ocean-area UGS has a low population density during normal times, the relative increase in their use during a pandemic may increase the risk of cross-infection. This could

exacerbate the rise in the number of cases and contribute to the positive correlation shown in the graph.

Nevertheless, we can see that the trend of increasing numbers of new cases began to plateau between October and December 2020 and after November 2021. At this point, the relative number of postings about ocean-area UGS began to rise rapidly and exceeded that of non-ocean-area UGS. This suggests that, after the effects of a pandemic are mitigated, the number of visitors to ocean-area UGS will rebound more quickly than the number of visitors to non-ocean-area UGS, reflecting strength in terms of tourism attractiveness. This is because, as the number of infected people declines, confidence and enthusiasm for tourism activities will gradually return. In addition, the government might respond to changes in unexpected circumstances in a timely manner by choosing to relax restrictions on tourism activities, such as by opening beaches and water recreation areas, which may also draw more visitors to ocean-area UGS.

The results also imply that non-ocean-area UGS has greater potential for tourism resource use and serves as a buffer against risk. Compared with the distinct seasonal usage pattern of ocean-area UGS due to climatic factors, non-ocean-area UGS has the advantage of being more flexible and adaptable as a tourism resource, is better able to cope with the impact of unexpected events, and avoids paralysis of the tourism sector. Non-ocean-area UGS, as an important part of urban infrastructure, can also function as an ecosystem service, providing a safer place for outdoor activities to meet public demand [42]. This is conducive to people's physical and mental health. In addition, in Okinawa, this type of green space is less affected by climatic factors and can provide a similar tourist experience all year round. As non-ocean-area UGS is easier to plan and renovate, it also offers a more diverse range of tourism programs (e.g., cultural, historical, and artistic events) to meet different tourism needs.

## 5. Conclusions

We used R to develop an ARIMA model of the number of postings about ocean-area UGS and non-ocean-area UGS in Okinawa. We aimed to compare the magnitude of the proportional change between the actual and predictive values after the outbreak and to analyze the seasonality of tourists' use of Okinawa's UGS and the impact of the COVID-19 pandemic.

The seasonality of the use of ocean-area UGS is dominated by climatic factors, with April–October being the peak season (especially during students' summer vacation in July), and November–March (continuing into the following year) being the off-peak season. The outbreak of the COVID-19 pandemic has had a significant impact on visitor enthusiasm, with a considerable reduction in the number of postings. Ocean-area UGS showed a greater variation in postings and a negative correlation with the number of new cases due to weather and safety concerns; a positive correlation was observed in the number of postings and the number of new cases for non-ocean-area UGS, demonstrating its ability to cope with risk during an outbreak.

Finally, the experience of the pandemic has brought about new demands for UGS management; in the post-pandemic era, with the spread of vaccines and changing perceptions of the pandemic, Okinawa's tourism industry now has opportunities to rebuild and recover. Restoring people's confidence in tourism will be a long-term and difficult task, specifically in terms of the following: (1) boosting the media buzz of under-recognized attractions and promoting a virtuous cycle of publicity; (2) making plans for the temporary renovation of UGS in response to unexpected social events or environmental weather crises; (3) making all data public in a timely manner and strengthening people's daily health and safety to bolster their confidence when traveling; and (4) promoting deep integration of the Internet, big data, artificial intelligence, and other relevant information technology with the tourism industry. Future prediction data could be used to rationalize services such as facility services, traffic management, and staff deployment to improve the quality of tourism services while avoiding wastage of resources. On the basis of the return of spontaneous tourists, we should actively advocate for people to resume social and outing

activities, reduce the psychological impact of the pandemic, and gradually work to remove the shadow of the pandemic.

This study has some limitations. The data source for building the ARIMA model was the only publicly available data source from the Tripadvisor platform; the accuracy of Internet data analysis is influenced by the frequency of Internet usage among visitors. However, it is crucial to acknowledge that there might be variations in Internet usage frequency across different age groups, regions with restricted access to the TripAdvisor platform, and economically disadvantaged areas. These variations can potentially result in statistical errors; more platforms could be considered to supplement the data source in the future. In addition, both the actual and predictive values are short-term data series. For future research on the "new normal" in the post-pandemic era, more data need to be added on an ongoing basis to gain a more comprehensive understanding of tourists' behavioral patterns. In addition, future studies should consider whether significant factors besides a pandemic influence the seasonality of visitor use; this could lead to more accurate predictions of visitor numbers.

**Author Contributions:** Conceptualization, R.Y. and J.Z.; methodology, R.Y.; software, R.Y.; validation, S.T. and J.Z.; formal analysis, K.L.; investigation, R.Y.; resources, S.L.; data curation, R.Y.; writing—original draft preparation, R.Y.; writing—review and editing, R.Y.; visualization, R.Y.; supervision, C.S.; project administration, R.Y.; funding acquisition, S.L. All authors have read and agreed to the published version of the manuscript.

**Funding:** This study was supported by a Grant-in-Aid for Scientific Research (No. 20K04884, Japan), JST SPRING (No. JPMJSP2109, Japan), and the Chinese Ministry of Education's Humanities and Social Science Project (No. 21YJCZH137, China).

**Informed Consent Statement:** Informed consent was obtained from all subjects involved in the study.

**Data Availability Statement:** All data used are openly accessible to the public.

**Acknowledgments:** The authors thank all authors for their advice and support for this study.

**Conflicts of Interest:** The authors declare no conflict of interest.

## Appendix A

This table presents the names of all the attraction sites and the cities involved in the UGC study data for this study.

**Table A1.** Attraction sites for the ocean-area UGS and the cities involved.

| Site | City | Site | City |
| --- | --- | --- | --- |
| Tomori Imugya Beach | Miyakojima | Ginowan Tropical Beach | Ginowan |
| Maehama Beach with Naha | Miyakojima | Ginowan Marina | Ginowan |
| Shinjo Coast | Miyakojima | Bibi Beach Itoman | Itoman |
| Shimodachi Island | Miyakojima | Odohama Beach | Itoman |
| Toriike Pond | Miyakojima | Kadeshi River | Itoman |
| Irabu-jima Island | Miyakojima | Nashiro Beach | Itoman |
| Ikema-jima Island | Miyakojima | Kawahira Bay | Ishigaki |
| Sawada no hama Beach | Miyakojima | Yonehara Beach | Ishigaki |
| Painagama Beach | Miyakojima | Sukuji Beach | Ishigaki |
| Kurimajima Island | Miyakojima | Shiraho Beach | Ishigaki |
| Watanabisama | Miyakojima | Sunset Beach | Ishigaki |
| Yae Ganse | Miyakojima | Maezato Beach | Ishigaki |
| Nakanoshima Beach | Miyakojima | Kabira Ishizaki Manta Scramble | Ishigaki |
| Shigira Beach | Miyakojima | Akashi Beach | Ishigaki |
| Ogami Island | Miyakojima | Maesato Beach | Ishigaki |
| Nagamahama Beach Coast | Miyakojima | Ishigaki-jima Blue Cave | Ishigaki |
| Sand Hill Beach | Miyakojima | Urasoko Bay | Ishigaki |

**Table A1.** *Cont.*

| Site | City | Site | City |
|---|---|---|---|
| Funakusu Beach | Miyakojima | Tomizaki Beach | Ishigaki |
| Hauai Waiwai Beach | Miyakojima | Osaki Hanagoi Reef | Ishigaki |
| Boraga Beach | Miyakojima | Osaki Tutle Reef | Ishigaki |
| Turiba Sunset Beach | Miyakojima | Iharama Okinone | Ishigaki |
| Hora Gyoko no Hama Beach | Miyakojima | Kumoji River | Naha |
| Kagimmi-hama Beach | Miyakojima | Ryutan | Naha |
| Maja Beach | Miyakojima | Naminoue Umisora Park | Naha |
| Opiiwa | Miyakojima | Miigusu Port | Naha |
| Yamatobu Oiwa | Miyakojima | Soongahinja Spring | Naha |
| Muigah Cliff | Miyakojima | Naminoue Beach | Naha |
| Tako Park | Miyakojima | Miigusuku Furusato Coast | Naha |
| Ikizu Beach | Miyakojima | Kudakajima Island | Nanjo |
| Satans Palace | Miyakojima | Ojima Island | Nanjo |
| Arasshisuhida Beach | Miyakojima | Miibaru Beach | Nanjo |
| Urasoko Beach | Miyakojima | Ojima Coast | Nanjo |
| Miyaguni Nnatohama Beach | Miyakojima | Komaka Island | Nanjo |
| Antoni Gaudi | Miyakojima | Azama Sunsun Beach | Nanjo |
| Muikaga | Miyakojima | Mibaru Beach | Nanjo |
| Nakanoshima Water Channel | Miyakojima | Ishiki Beach | Nanjo |
| Kumaza Beach | Miyakojima | Habyan, Cape Kaberu | Nanjo |
| Nagakita Beach | Miyakojima | Ukabijima Island | Nanjo |
| Cross Hole | Miyakojima | Pizza Beach | Nanjo |
| Miyako Sunset Beach | Miyakojima | Busena Beach | Nago |
| Hamahiga Island | Uruma | 21st Century Forest | Nago |
| Ikei Beach | Uruma | Yagaji Island | Nago |
| Odomari Beach | Uruma | Kise Beach | Nago |
| Tsukenjima Island | Uruma | Kanucha Beach | Nago |
| Tonnaha Beach | Uruma | Nago citizen Beach | Nago |
| Muruku Hama Beach | Uruma | Sea Glass Beach | Nago |
| Hamahiga Beach | Uruma | Koki Beach | Nago |
| MIyagijima Island | Uruma | Yagaji Beach | Nago |
| Henza Island | Uruma | Teniya Beach | Nago |
| Tsukenjima Beach | Uruma | Setagashima | Tamagusuku |
| Tomai-hama Beach | Uruma | Toyosaki Kaihin Koen | Tamagusuku |
| Ukibaru-jima Island | Uruma | Chura Sun Beach | Tamagusuku |
| Itsukuma Beach | Uruma | Hija River | Okinawa City |
| Kaneku Beach | Uruma | | |

**Table A2.** Attraction sites for the non-ocean-area UGS and the cities involved.

| Site | City | Site | City |
|---|---|---|---|
| Miyakojima Marine Park | Miyakojima | Ishigaki Island Science Garden | Ishigaki |
| Imugya Marine Garden | Miyakojima | Hirakubo Sagaribana Gunraku | Ishigaki |
| Shimajiri Mangrove Forests | Miyakojima | Maezato Dam | Ishigaki |
| Hirara Tropical Botanical Garden | Miyakojima | Oura Dam | Ishigaki |
| Hika Road Park | Miyakojima | Kanmuriwashi Observatory | Ishigaki |
| Turiba Seaside Park | Miyakojima | Urasoe Park | Urasoe |
| Nishikaigan Park | Miyakojima | Urasoe Sports Park | Urasoe |
| Kamamamine Park | Miyakojima | Miyagi Park | Urasoe |
| Shiratori Misaki Park | Miyakojima | Ohira Bus Stop Park | Urasoe |
| Nakahara Limestone Cave | Miyakojima | Fukushuen | Naha |
| Stone Garden | Miyakojima | Shikinaen | Naha |
| Tropical Fruits Park | Miyakojima | Onoyama Park | Naha |

**Table A2.** *Cont.*

| Site | City | Site | City |
| --- | --- | --- | --- |
| Takenakayama Tembo Park | Miyakojima | Shurikinjocho Oakagi Tree | Naha |
| Fukuzato Underground Dam | Miyakojima | Manko Park | Naha |
| Minafuku Underground Dam Park | Miyakojima | Yogi Park | Naha |
| Painagama Umizora Sukoyaka Park | Miyakojima | Sueyoshi Park | Naha |
| Ogamijima Island Multipurpose Park | Miyakojima | Wakasa Seaside Park | Naha |
| Shiratori Hole | Miyakojima | Kinjo Dam | Naha |
| Panata | Miyakojima | Matsuyama Park | Naha |
| Sunken Ship Irabu | Miyakojima | Midorigaoka Park | Naha |
| Bios Valley | Uruma | Kibogaoka Park | Naha |
| Cave Okinawa | Uruma | Asatogawa Shinsui Park | Naha |
| Zukeran Poultry Farm Minimini Zoo | Uruma | Uenomo Park | Naha |
| Jyane Caves | Uruma | Gajanbira Park | Naha |
| Sea Side Garden Hamahiga | Uruma | Makishi Park | Naha |
| Miten Uza | Uruma | Asahigaoka Park | Naha |
| Kurashiki Dam | Uruma | Shintoshin Park | Naha |
| Uruma Shiminnomori Park | Uruma | Okinawa Cellular Park Naha | Naha |
| Yacho no Mori Nature Park | Uruma | Kuganimui Park | Naha |
| Iha Park | Uruma | Matsuo Park | Naha |
| Heshikiya Takino | Uruma | Uenoya North Park | Naha |
| Hamagyoko Ryokuchi Park | Uruma | Sakiyama Park | Naha |
| Hanaridaki | Uruma | Nami no Ue Chocho House | Naha |
| Ginowa Seaside Park | Ginowan | Ai no Shisa Park | Naha |
| Kakazu Upland Park | Ginowan | Gangala Valley | Nanjo |
| Morikawa Park | Ginowan | Cape Chinen Park | Nanjo |
| Mashiki Pocket Park | Ginowan | Hanayakamura | Nanjo |
| Heiwa Sozo no Mori Park | Itoman | Gusuku Road Park | Nanjo |
| Okinawa Maha Bodhi Garden | Itoman | Chichinga | Nanjo |
| Ishigaki Island Stalactite Cave | Ishigaki | Busena Marine Park | Nago |
| Ishigakijima Banner Park | Ishigaki | NEO PARK OKINAWA | Nago |
| Kawahira Park | Ishigaki | Forest Yanbaru Subtropical | Nago |
| Ibaruma Sabichi Cave | Ishigaki | Nago Castle Historical Park | Nago |
| Nosoko Mape | Ishigaki | Todoroki Falls | Nago |
| Yaeyama Shyonyudo Doshokubutsuen Park | Ishigaki | 21st Century Forest Park | Nago |
| Nosoko Forest Road | Ishigaki | Mt. Tanoudake | Nago |
| Shinei Park | Ishigaki | Fukugawa Falls | Nago |
| Misaki Park | Ishigaki | Haneji Dam | Nago |
| Maezato Park | Ishigaki | Shikuwasa Hana & Green Village | Nago |
| Funakura Park | Ishigaki | Kouki Park | Nago |
| Sokobaru Dam | Ishigaki | Kaigungo Park | Tamagusuku |
| Mr. Isigaki Garden | Ishigaki | Manko Waterbird & Wetland Center | Tamagusuku |
| Ishigaki Dam | Ishigaki | Dmm Kariyushi Aquarium | Tamagusuku |
| Arakawa Falls | Ishigaki | Southeast Botanical Gardens | Okinawa City |
| Nagura Dam | Ishigaki | Okinawa Zoo & Museum | Okinawa City |
| Kids Land Fantasy World | Ishigaki | Okinawa Comprehensive Athletic Park | Okinawa City |
| Manta Park | Ishigaki | Yaeshima Park | Okinawa City |
| Yashima Ryokuchi Park | Ishigaki | Akemichi Park | Okinawa City |
| Yashima Jinko Island | Ishigaki | | |

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
