# Peer review of "Quantitative Analysis of Seasonality and the Impact of COVID-19 on Tourists’ Use of Urban Green Space in Okinawa: An ARIMA Modeling Approach Using Web Review Data"

_land, doi:10.3390/land12051075_

Round 1

Reviewer 1 Report

Introduction: The last paragraph highlights detail part of the method which should not be under Introduction. Bring it to your method section

Method:

For 2.1 please provide a map contains Okinawa and the study sites. A map will make the readers easy to find where is Okinawa on the map.

Author Response

 We would like to thank you for your careful and thorough reading of this manuscript and for the thoughtful comments and constructive suggestions, which have helped to improve the quality of this manuscript.

Please refer to the accompanying response document (Response_to_Reviewer_1_.pdf) and the revised version (land-2331477.pdf / land-2331477.doc) for specific changes. All page and line numbers refer to the revised manuscript file.

Reviewer 2 Report

The perspective of creating predictive models using user-generated data is very interesting.

However, the advantages of user-generated data are not fully exploited. First of all, why did you use user-generated data, the number of posts on TripAdvisor, instead of the exact number of visitors?

How many visitors to Okinawa use TripAdvisor? The legitimacy of choosing TripAdvisor must be proven through specific statistics and comparisons of SNS used by Japanese visitors.

Several topics are being used in the present thesis. A clear definition of the subject is required.

Author Response

We would like to thank you for your careful and thorough reading of this manuscript and for the thoughtful comments and constructive suggestions, which have helped to improve the quality of this manuscript.

Please refer to the accompanying response document (Response_to_Reviewer_2_.pdf) and the revised version (land-2331477.pdf / land-2331477.doc) for specific changes. All page and line numbers refer to the revised manuscript file.

Reviewer 3 Report

This paper analyses the impact of Covid 19 on tourist visitation to the land and sea at Okinawa, Japan covering the period, January, 2014 to December, 2021. Monthly records of visitation were obtained from the online app, Trip Advisor, and analysed for the land and sea. A sophisticated tool, autoregressive integrated moving average (ARIMA), was used in the analysis together with a range of other tools, including Beautiful Soup, Scrapy crawler and PyMongo.

Based on the data for the pre-pandemic period, 2014 to December, 2019, projections were made for the pandemic period, January, 2020 to December, 2021 for both the sea and land areas. Interestingly, the data showed that the visitation for both areas declined from around 2016 to 2019, moreso for the land area than the sea. The project analysed the difference between the projected trend and the actual data for the pandemic period. The sea area experienced greater variation in Trip Advisor postings and these were negatively correlated with the number of new cases of Covid-19. However, for the land area, there was a positive correlation between new Covid-19 cases and postings, leading to the conclusion that the land area was more resilient in the face of unexpected events. Following Covid-19 however, postings for the sea area bounced back more strongly than the land area. In both areas, visitation is strongly influenced by seasonal climate (i.e. summer, winter) and by the availability of public holidays.

Strengths of the paper include the clear description of its method, analytical tools and data sources, the thoroughness of its analysis and the clarity of its tables and figures. The paper includes suggestions to strengthen management of the tourist resources in Okinawa as they recover from the pandemic. The English is excellent and no faults were detected.

A weakness is the use of Trip Advisor as the primary source of visitor data. Not everyone who uses Trip Advisor actually go to the destination. It is often used to evaluate a range of possible locations. This aspect was not discussed in the paper. Actual visitor data could have been obtained through instruments such as Booking.com, Trivago.com, Hotels.com, Agoda.com, Expedia.com and similar on-line booking platforms. There are also probably specialised Japanese booking platforms available.

A map delineating the ocean and non-ocean parts of the study would be useful.

Minor correction

Fig 16 This shows ocean area UGS and non-ocean area UGS but the legend to the figure lists both lines as Ocean area urban green space.

Author Response

We would like to thank you for your careful and thorough reading of this manuscript and for the thoughtful comments and constructive suggestions, which have helped to improve the quality of this manuscript.

Please refer to the accompanying response document (Response_to_Reviewer_3_.pdf) and the revised version (land-2331477.pdf / land-2331477.doc) for specific changes. All page and line numbers refer to the revised manuscript file.

Round 2

Reviewer 3 Report

I have reviewed the changes the authors have provided in this paper. They have made many editorial changes throughout the paper and have deleted certain paras and sentences and added substitutes. They have included a map of the location of the cities in Okinawa and have also added two appendixes which list the cities in the ocean area UGS and non-ocean area UGS. These are all useful additions and changes.
I consider the paper sufficiently improved to warrant publication in the Land journal.

Author Response

Thank you very much for your hard work in reviewing this paper.